# Phosphoproteomic Analysis Identifies TYRO3 as a Mediator of Sunitinib Resistance in Metastatic Thymomas

**DOI:** 10.3390/cancers14194762

**Published:** 2022-09-29

**Authors:** Stefan Küffer, Jessica Grabowski, Satoru Okada, Nikolai Sojka, Stefan Welter, Alexander von Hammerstein-Equord, Marc Hinterthaner, Lucia Cordes, Xenia von Hahn, Denise Müller, Christian Sauer, Hanibal Bohnenberger, Alexander Marx, Philipp Ströbel

**Affiliations:** 1Institute of Pathology, University Medical Center Göttingen, University of Göttingen, 37075 Göttingen, Germany; 2Division of Thoracic Surgery, Department of Surgery, Graduate School of Medical Science, Kyoto Prefectural University of Medicine, Kyoto 602-8566, Japan; 3Thoracic Surgery Department, Lung Clinic Hemer, 58675 Hemer, Germany; 4Department of Thoracic and Cardiovascular Surgery, University Medical Center, 37075 Göttingen, Germany; 5Institute of Pathology, University Medical Center Mannheim, University of Heidelberg, 68167 Mannheim, Germany

**Keywords:** phosphoproteomics, sunitinib, TYRO3, drug response, biomarker, thymoma, thymic carcinoma, tyrosine kinase

## Abstract

**Simple Summary:**

After initially responding to empiric radio-chemotherapy, most advanced thymomas and thymic carcinomas become refractory and require second-line therapies. The multi-target tyrosine kinase inhibitor, sunitinib, is one of few options, especially in patients with thymic carcinomas, and has resulted in partial remissions and prolonged overall survival. However, sunitinib shows limited activity, and not all patients benefit equally. A better understanding of its mode of action and the definition of predictive biomarkers would help select patients who profit most. Using a real-time multiplex tyrosine phosphorylation assay containing 144 kinase substrates in a defined set of sunitinib-sensitive and -resistant cell lines, we generated a sunitinib response index (SRI). Protein lysates from thymomas and thymic carcinomas with spike-in of sunitinib were then classified as potential responders vs. non-responders using the same SRI classifier. Bioinformatic prediction and further experimental analysis of activated upstream kinases identified TYRO3 as a potent mediator of sunitinib resistance, specifically in metastatic thymomas. TYRO3 could serve both as a biomarker of sunitinib resistance and a potential therapeutic target that could help to tailor treatment decisions and to overcome therapy resistance in advanced thymomas and thymic carcinomas.

**Abstract:**

Background: After initially responding to empiric radio-chemotherapy, most advanced thymomas (TH) and thymic carcinomas (TC) become refractory and require second-line therapy. The multi-target receptor tyrosine kinase (RTK) inhibitor, sunitinib, is one of the few options, especially in patients with thymic carcinomas, and has resulted in partial remissions and prolonged overall survival. However, sunitinib shows variable activity in thymomas, and not all patients benefit equally. A better understanding of its mode of action and the definition of predictive biomarkers would help select patients who profit most. Methods: Six cell lines were treated with sunitinib in vitro. Cell viability was measured by MTS assay and used to define in vitro responders and non-responders. A quantitative real-time assay simultaneously measuring the phosphorylation of 144 tyrosine kinase substrates was used to correlate cell viability with alterations of the phospho-kinome, calculate a sunitinib response index (SRI), and impute upstream tyrosine kinases. Sunitinib was added to protein lysates of 29 malignant TH and TC. Lysates were analyzed with the same phosphorylation assay. The SRI tentatively classified cases into potential clinical responders and non-responders. In addition, the activation patterns of 44 RTKs were studied by phospho-RTK arrays in 37 TH and TC. Results: SRI application separated thymic epithelial tumors (TET) in potential sunitinib responders and resistant cases. Upstream kinase prediction identified multiple RTKs potentially involved in sunitinib response, many of which were subsequently shown to be differentially overexpressed in TH and TC. Among these, TYRO3/Dtk stood out since it was exclusively present in metastatic TH. The function of TYRO3 as a mediator of sunitinib resistance was experimentally validated in vitro. Conclusions: Using indirect and direct phosphoproteomic analyses to predict sunitinib response in malignant TET, we have shown that TH and TC express multiple important sunitinib target RTKs. Among these, TYRO3 was identified as a potent mediator of sunitinib resistance activity, specifically in metastatic TH. TYRO3 may thus be both a novel biomarker of sunitinib resistance and a potential therapeutic target in advanced thymomas and thymic carcinomas.

## 1. Introduction

Thymomas (TH) and thymic carcinomas (TC) are rare epithelial tumors of the thymus. According to the World Health Organization (WHO) classification of thymic epithelial tumors (TET) [1,2], TH are subclassified into types A, AB, B1, B2, and B3. While most TH have organotypic features, the more aggressive TC are morphologically indistinguishable from carcinomas in other organs [3]. Among TH, type B2 and B3 frequently show invasion of neighboring organs, pleural dissemination, and even hematogenous metastases. Surgery with complete tumor resection is the only curative approach [4]. Unresectable and metastatic TET require chemotherapy in combination with radiotherapy [5]. However, therapy failure is frequent, and many patients will be candidates for second-line treatment. Unfortunately, TET are among the adult tumors with the lowest mutational burden, and TET patients have hardly benefited from precision medicine approaches [6]. Therefore, targeting activated receptor tyrosine kinases may be a promising approach in these “non-mutated” tumors. We have previously described a TC with mutated *KIT* and partial response to imatinib [7]. However, *KIT* mutations occur in less than 10% of TC and do not occur in TH [6]. Cetuximab shows some activity in TET overexpressing non-mutated epidermal growth factor receptor (EGFR) [8,9,10], but its overall efficacy is unsatisfactory. Previous reports described long-lasting partial remissions in patients with metastatic TC treated with the multi-kinase inhibitor sunitinib [11], and Thomas et al. showed a high response rate in pretreated patients with TC in an open-label phase 2 trial [12]. According to NCCN guidelines (https://www.nccn.org/), sunitinib is now one of the few recommended second-line therapeutic options for treating TC. Although Thomas et al. found no advantage in advanced TH [12], data from the French RYTHMIC consortium showed similar overall response rates for TH and TC [13]. These data indicate that not all patients benefit equally from sunitinib, making a better understanding of its specific mode of action in TET, and, ideally, a predictive biomarker desirable. However, the very broad activity of sunitinib [14], the rarity of TET, and the limited availability of tumor tissue from patients treated with sunitinib likely make this mission impossible. We tried to overcome this impasse through an indirect approach: a panel of sunitinib-sensitive and -resistant cell lines was used to define resistance patterns using quantitative profiling of tyrosine kinase substrates and to calculate a sunitinib response index (SRI). This chip-based method measures the phosphorylation of 144 tyrosine kinase substrates by the kinase activity of the analyzed cell or tissue lysate in real time. Since different tyrosine kinases have different phosphorylation targets, bioinformatic tools can predict the upstream tyrosine kinase that has likely caused a given phosphorylation pattern. By comparing two lysates of the same sample with and without the addition of a tyrosine kinase inhibitor (such as sunitinib), the direct effect of the inhibitor on the phosphorylation pattern of the substrates (and thus on the kinase) can be quantified. The same method and the SRI generated and trained in vitro were then used to analyze and tentatively classify protein lysates of 29 malignant TET into potential sunitinib responders and non-responders. Further analysis revealed activated TYRO3/Dtk as a potential biomarker of TET with more aggressive behavior and resistance towards sunitinib. 

## 2. Material and Methods

### 2.1. Clinical Patient Data and Tissues

TET samples were classified according to the recent WHO classification of 2021 by expert pathologists (AM and PS). The histological tumor stage was assessed according to the modified Masaoka–Koga classification [15]. Native specimens were snap-frozen in liquid nitrogen and stored at −80 °C until further processing. All procedures were performed in accordance with the seventh version of the Declaration of Helsinki (General Assembly of the World Medical 2014) The project was approved by the ethics committee of the University Medical Center Göttingen (GÖ 912/15) (Table 1).

### 2.2. Cell Culture

PC-3, HCC15, NTERA-2, MCF-7, and LNCaP cells were purchased from DSMZ-German Collection of Microorganisms and Cell Cultures GmbH. The human TC cell line 1889c was kindly provided by Ehemann et al. [16]. Cells lines were cultured in RPMI-1640 medium supplemented with 10% fetal bovine serum, 2 mM L-glutamine, and 100 U/mL penicillin/streptomycin (Gibco, Waltham, MA, USA) at 37 °C in a 5% CO_2_ humidified environment.

### 2.3. Lysis of Tissues and Cells and Protein Extraction

Cryopreserved tissues and fresh cells were lysed using ice-cold M-PER lysis buffer (Thermo Fisher Scientific, Waltham, MA, USA) containing a 1:100 Protease Inhibitor Cocktail (Thermo Fisher Scientific, Waltham, MA, USA). The cell lysate was centrifuged at 14,000× *g* for 10 min at 4 °C and the supernatant was snap-frozen in aliquots and stored at −80 °C. The protein concentration was determined by the BCA Assay using the Pierce™ bovine serum albumin standard (Thermo Fischer Scientific, Waltham, MA, USA).

### 2.4. Kinase Activity Profiling on PamChip^®^ Peptide Microarrays 

Kinase activity profiles were measured using the PamChip^®^ 12 protein tyrosine (PTK) peptide microarray on a Pamstation 12 system from PamGene (PamGene International B.V., ‘s-Hertogenbosch, The Netherlands) according to the manufacturers’ recommendations and as described before [17,18]. In brief, microarrays were blocked with 2% BSA and then washed three times with PK buffer before 2 µg cell lysate or 5 µg tissue lysate were applied in a 40 µL mixture containing 50 mM Tris-HCl pH 7.5, 10 mM MgCl2, 1 mM EGTA, 2 mM dithiothreitol, 0.01% Brij-35, 1 mg/mL BSA, 12.5 μg/mL FITC-labeled PY-20 antibody, and 0.4 mM ATP (PamGene). Fluorescent images were taken every 5 cycles over 62 cycles. Each sample was measured with 5 µM sunitinib spike-in and DMSO as control. 

### 2.5. Image Filtering, Data Adaptation and Prediction Model Generation

For sample and array annotation, image gridding, quality control (QC), and quantification of the phosphorylation signal, the software package BioNavigatoR (version 6.2; PamGene, ‘s-Hertogenbosch, The Netherlands) was used. The slope of the peptide signal intensity across different exposure times (10, 20, 50, 100, and 200 ms) was multiplied by 100 and log2-transformed. Sunitinib inhibition was defined as the log2 ratio of the signal obtained in the same sample treated with 5 µM sunitinib vs. DMSO control. The inhibition ratios were used for all subsequent calculations. The SRI prediction classifier based on six different responding cell lines was established using Partial Least Squares Discriminant Analysis (PLS-DA). This method has previously been used to classify clinical samples and to rank the best-performing classifier with PamChip^®^ peptide microarray data [19,20]. PLS-DA was applied without selecting discriminative peptides or discriminating classes of interest.

### 2.6. Sunitinib Response Upstream Kinase Prediction

To predict kinases involved in the sunitinib resistance, tyrosine phosphorylation sites of a two-group comparative analysis between resistant and non-resistant samples (quantifier and predicted tissue samples) were subjected to upstream kinase prediction using the upstream kinase operator in the BioNavigatoR software (Version 6.2; PamGene, ‘s-Hertogenbosch, The Netherlands). Kinases known to phosphorylate specific peptides were identified in databases and kinases most likely to influence the phosphorylation pattern were permutated for sensitivity and specificity [21].

### 2.7. Transfection of siRNA and Expression Plasmids and Cell Viability Measurement

Cells were reverse transfected with either plasmid DNA or siRNA. Plasmid transfection was performed using the X-tremeGENE HP DNA transfection reagent (Merck, Germany) according to the manufacturer’s instructions. In brief, 100 µL transfection mix containing serum-free RPMI-1640 cell culture medium, 2 µg plasmid DNA, and 2 µL transfection reagent was incubated for 15 min at room temperature and directly added to 4 × 10^5^ cells in 2 mL of medium after seeding. The TYRO3 and the control expression vectors were purchased from OriGene, USA and Thermofisher, USA (TYRO3:pCMV6-XL4, CAT#: SC108283, pcDNA^TM^3.1(+), CAT#: V790-20). For siRNA transfection, HiPerFect (Qiagen, Germany) was used according to the manufacturer’s protocol. Briefly, a transfection mix of 100 µL serum-free RPMI-1640, 100 nM siRNA, and 12 µL transfection reagent was incubated for 5 min and added to 4 × 10^5^ cells in 2.3 mL medium and incubated for 24 h before treatment. siRNAs were purchased from Qiagen, Germany (SI00288344, SI00288351). Cell viability was measured using the CellTiter 96^®^ AQueous One Solution Cell Proliferation Assay (Promega, Madison, WI, USA) according to the manufacturer’s recommendations. Absorbance was measured using a Tecan Plate Reader 2000 (Tecan, Switzerland). 

### 2.8. Protein Extraction and Western Blot

Protein isolation from cells was performed using RIPA lysis buffer containing 1x protease inhibitor cocktail Complete (Roche, Basel, Switzerland), 1 mM PMSF, and 1 mM orthovanadate (Sigma-Aldrich, St. Louis, MO, USA). The protein concentration was determined using DC™ Protein Assay (Bio-Rad, Feldkirchen, Germany). Western blots were performed using precast Mini Protein TGX gels and the semi-dry Trans-Blot Turbo^TM^ System (Bio-Rad, Germany). Antibodies and related secondary antibodies (DAKO, Tokyo, Japan) were used at a dilution of 1:1000 in TBST for Anti-Tyro3 (Cell Signalling, Danvers, MA, USA, #5585). Anti-GADPH (Cell Signalling, USA, #5174) was used as a loading control.

### 2.9. Protein Extraction and Screening for Activated RTKs

Protein lysates for the Proteome Profiler^TM^ Array (Human Phospho-RTK Array Kit) were prepared as previously described (Strobel, et al. 2010, [11]). In brief, 15 5 µm sections of fresh, frozen tissue sections were lysed in 1 mL Lysis Buffer 17 and subjected to 10 μg/mL Aprotinin, 10 μg/mL leupeptin, and 10 μg/mL pepstatin at 4 °C for 30 min before centrifugation at 14,000× *g* for 5 min. The protein quantity of the supernatant was determined using Pierce 660 nm Protein Assay Reagent (Thermo Fisher Scientific, Waltham, MA, USA). The arrays were incubated overnight with 250 µg of protein at 4 °C. Membranes were washed twice with wash buffer before incubating with an anti-phosphotyrosine-HRP detection antibody for 2 h at room temperature. For imaging, the chemiluminescent detection reagent from the array kit was mixed in at a ratio of 1:1. After two additional wash steps, the membrane was incubated for 1 min with 1 mL of the reagent mix, and imaging was performed using the Fusion FX7 chemiluminescence detection system. The signal intensity was analyzed using the software ImageJ V1.51 (NIH, New York, NY, USA).

## 3. Results

### 3.1. RTK Multiplex Tyrosine Phosphorylation Assay of Six Cell Lines and Ex Vivo TH and TC Samples and Generation of the SRI

To establish a general model for sunitinib resistance, we used six cell lines from different entities, including the TC cell line 1889c, the breast cancer cell line MCF-7, the prostate cancer cell lines PC3 and LNCaP, the teratoma cell line NTERA-2 (NT2), and the lung squamous cancer cell line HCC15. Except for 1889c, the cell lines had been selected based on their IC50 for sunitinib provided by the Welcome Sanger Institute Database “Genomics of Drug Sensitivity in Cancer” (https://www.cancerrxgene.org/compound/Sunitinib/5/overview/ic50, accessed on 1 September 2022). Cells were subjected to 5 µM sunitinib for 48 h, and viability was measured by MTS. NT2 showed the best response, and PC3 was the most resistant to sunitinib (Figure 1a). We then used protein lysates from all cell lines and from 29 fresh, frozen malignant TH and TC for real-time quantification of phosphorylated tyrosine residues as described in the material and methods section (Appendix A). Each lysate was analyzed twice with and without adding 5 µM sunitinib (Appendix A). The resulting ratios of the six cell lines were analyzed by unsupervised clustering (Figure 1b). By combining the in vitro cell survival data (defining sunitinib responsive and resistant cells) and the phosphorylation patterns, we next calculated a prediction classifier termed sunitinib response index (SRI) (Figure 1c). To identify the kinases that were potentially responsible for the observed patterns, we performed a bioinformatic upstream kinase prediction. The 21 top-ranked tyrosine kinases are shown in Figure 1d (a complete list of identified kinases is provided in Appendix A). The predicted kinases contained several known sunitinib targets (in particular FLT3, FLT4, Mer, Axl, and TYRO3) [22]. 

### 3.2. The SRI Applied to Clinical TET Tissue Samples Predicts Differential Response to Sunitinib 

We next applied the SRI to protein lysates from 29 malignant TH and TC. This resulted in two distinct groups of potential sunitinib responders and non-responders (Appendix A). Interestingly, most TC were potential responders in line with clinical observations, whereas TH were more heterogeneous. In addition, the signal ratio of sunitinib vs. DMSO control was higher in TC (Appendix A). Common predicted kinases were ROR1, ROR2, TYRO3, and FRK (Appendix A). We then analyzed TH and TC separately. In TC, hierarchical clustering showed two main groups with 3 out of 10 samples (30%) with high ratios (Figure 2a). These three cases also showed the highest SRI values (Figure 2b). The top-ranked kinases were predicted with high specificity scores and again included known sunitinib targets, particularly BLK, Yes, Lyn, Lck, PDGFRA, ABL, and INSR. 

The 15 predicted kinases with known sunitinib binding included *n* = 8 (53%) with high selectivity (quantitative dissociation constant, K_d_ < 1 μM) (Davis, et al. 2011) (Figure 2c and Appendix A). Of note, a primary sunitinib target, KIT, was not predicted, indicating that KIT is functionally relevant only in those few TC with activating KIT mutations (Strobel, et al. 2004).

In TH, the signal ratio of sunitinib vs. DMSO control was more evenly distributed (Figure 2d). Among the 19 TH samples, 16 were predicted as potential sunitinib responders, although only 3 cases (15.8%) reached high SRI values (Figure 2e). Top-ranked predicted upstream tyrosine kinases that were also primary sunitinib targets included TYRO3/Dtk, ITK, ABL, and TRKB (Figure 2f and Appendix A). The 11 predicted kinases with known sunitinib binding included *n* = 5 (45%) (TYRO3/Dtk, FRK, ITK, ABL, TRKB) with a K_d_ < 1 μM. Among those, only ABL overlapped with TC, and only TYRO3/Dtk overlapped with cell lines. 

### 3.3. Phospho-RTK Arrays in Clinical TET Samples Reveal Stage-Related Activation Patterns of EGFR and TYRO3/Dtk

To better understand the obviously different resistance mechanisms towards sunitinib in TH and TC, we analyzed snap-frozen tissue samples of *n* = 7 TC and *n* = 30 TH using a commercially available array for the measurement of 44 phosphorylated (activated) RTKs (Figure 3). Among the 44 RTKs represented on the array, and with published data on sunitinib binding, were 22 with high (K_d_ < 1 μM) and 11 with low (K_d_ > 3 μM) selectivity for sunitinib. Interestingly, there was no clear-cut separation based on histology (TH vs. TC). In TC, the most frequently activated RKT was EGFR (a low-affinity target of sunitinib with a K_d_ > 3 μM), and KIT was not activated in any of the TC samples studied here (all negative for KIT mutations). The most frequently activated RTKs that also bind sunitinib with high selectivity (K_d_ < 1 μM) were (in descending order) EphA3, IGF1R, AXL, EphA6, FGFR3, FGFR4, and VEGFR3. In TH, there was a striking and nearly exclusive dichotomy between primary and metastatic tumors. While 17/23 of primary TH (74%) showed strong activation of EGFR, all 7 metastatic TH (100%) were negative for EGFR, and 4 out of 7 (57%) showed activation of TYRO3/Dtk instead (Figure 3a). There was not a single case with activation of both EGFR and TYRO3/Dtk. Upon quantification, TYRO3/Dtk was significantly more phosphorylated in recurrent and metastatic TH than in primary TH and TC (Figure 3b), while the EGFR was significantly more phosphorylated in primary TH and TC (Figure 3c). All other activated RTKs did not differ significantly between TH and TC. The most frequently activated RTKs, other than TYRO3/Dtk, that also bind sunitinib with high selectivity (K_d_ < 1 μM) in TH were (in descending order): FGFR2 (*n* = 11 cases), VEGFR3 (*n* = 9 cases), TRKB and EphA6 (*n* = 8 cases each), VEGFR1 (*n* = 7 cases), EphB6, FGFR4, FLT3, INSR, and VEGFR2 (*n* = 6 cases each). Of note, there were also three TH cases with activation of SCFR/KIT. The most frequently activated RTKs other than EGFR that bind sunitinib with low selectivity (K_d_ > 3 μM) in TH were (in descending order): Tie-2 (*n* = 10), TRKC (*n* = 9), ERBB3, and ERBB2 (*n* = 4 cases each).

### 3.4. TYRO3/Dtk Activity Correlates with Sunitinib Response in Cell Lines

After testing pTYRO3/Dtk activity in the six cell lines, we detected variable signals from activated TYRO3 (Figure 4a,b). The pTYRO3/Dtk signal intensity was statistically significantly correlated with sunitinib response (Figure 4c). To further validate the functional importance of TYRO3/Dtk, we used an expression vector to transiently overexpress TYRO3/Dtk in the two cell lines with the lowest constitutive TYRO3 activation and the best response to sunitinib (MCF-7, and NT2). This significantly increased the resistance against sunitinib in both cell lines (Figure 4d,f). Vice versa, in the four cell lines with strong constitutive activation of TYRO3/Dtk (1889c, PC3, HCC15, and LNCaP), gene silencing with two individual siRNAs resulted in significantly more tumor cell killing upon exposure to sunitinib (Figure 4e,g). 

## 4. Discussion

Sunitinib is a potent multi-tyrosine kinase inhibitor targeting numerous tyrosine kinases, including KIT, vascular endothelial growth factors 1-3 (VEGFR 1-3), FMS-like tyrosine kinase 3 (FLT3), platelet-derived growth factors (PDGFRA and PDGFRB), and colony-stimulating factor 1 (CSF1) [14,23,24]. It is used as a second-line treatment in advanced pretreated TET [11,12,13]. However, the available clinical data show variable response rates, especially in TH, making predictive biomarkers highly desirable. Sunitinib’s mode of action and even the presence of its primary target RTKs in TET have not been studied in detail before. The very limited availability of tissue samples of TET patients treated with sunitinib and the broad activity of sunitinib virtually preclude straight-forward functional studies. Here we describe an indirect strategy to overcome this problem using quantitative phosphoproteomics to measure tyrosine kinase targets in a panel of sunitinib-sensitive and -resistant cell lines to calculate an SRI. The same method and the SRI were then used to analyze and classify 29 malignant TET into potential responders and non-responders.

The validity of this approach was illustrated by the fact that the predicted tyrosine kinases included numerous prominent sunitinib targets and helped to identify TYRO3/Dtk as a major resistance factor in metastatic TH. Our results show for the first time that the mechanisms determining sensitivity towards sunitinib are very likely different in TH and TC. In line with clinical observations, our model predicted more potential sunitinib responders among TC than among TH (30% in TC vs. 16% in TH). In TC, more than half (53%) of the predicted upstream tyrosine kinases involved in sunitinib response had very high selectivity (K_d_ < 1 μM) for sunitinib [22]. Notable examples included BLK, Yes, Lyn, Lck, PDGFRA, ABL, and INSR. On the other hand, when analyzing which RTKs were actually expressed in TET, the most constantly activated RTK was EGFR, which is not a sunitinib target. In addition, KIT was neither predicted by the model nor was it among the RTKs found activated in TC tissue lysates—even though KIT (CD117) is probably the most constantly expressed immunohistochemical marker in TC (Pan, et al. 2004). This finding is a direct explanation for the disappointing results of a clinical trial studying the efficacy of imatinib in unselected TC patients [25] as opposed to impressive responses in TC patients with gain of function *KIT* mutations [7,26,27,28,29]. The most frequently activated RTKs in TC that are also known as sunitinib targets included EphA3, EphA6, IGF1R, AXL, FGFR3, FGFR4, and VEGFR3. 

In TH, 45% of the predicted upstream kinases were high-affinity sunitinib targets with a K_d_ < 1 μM and included TYRO3, FRK, ITK, Abl, and TRKB. An analysis of activated RTKs in TH tumor lysates revealed a striking dichotomy between localized and metastatic tumors with a switch from EGFR to activated TYRO3 in 57% of metastatic cases. This finding has potential clinical significance since patients with metastatic TH are the most likely candidates for second-line treatment with sunitinib. The crucial role of TYRO3 as a mediator of sunitinib resistance in TH was further experimentally substantiated by overexpression and silencing in tumor cell cultures. TYRO3 belongs to the TAM (TYRO3-ABL-MER) family of RTKs, which can regulate tumor cell survival, proliferation, and angiogenesis (reviewed in [30]). Notably, a large body of evidence links the members of this family to resistance both towards targeted therapies and conventional chemotherapy [30]. Early clinical data evaluating the selective TAM-kinase inhibitor, sitravatinib (MGCD516), in advanced solid cancers have shown manageable safety and modest clinical activity [31]. TYRO3 is the least studied member of the TAM family and has not been described in connection with sunitinib. Knockdown of TYRO3 has been shown to suppress the growth of myeloid leukemia cells [32]. Its functions appear to partially overlap with the other members of the TAM family and involve cell cycle progression and anti-apoptosis via MAPK/ERK and PIK3/AKT signaling pathways. 

Apart from TYRO3, the most frequently activated RTKs that also bind sunitinib with high selectivity in TH were FGFR2, FGFR4, VEGFR 1-3, TRKB, FLT3, INSR, and EphA6 and EphB6. Ephrin receptors were also frequently activated in TC. They constitute the largest subfamily of RTK. A large body of data has linked Ephrin receptors to a multitude of oncogenic key events, including angiogenesis, metastasis, and perineural and vascular invasion [33]. Frequent expression of ephrin receptors and EphA6, in particular, has been pointed out earlier in epithelial-rich TET [34]. Interestingly, there is evidence of a synergy between the EGFR (the most frequently activated RTK in our study) and ephrin receptors in cancer progression [35]. For example, blockade of EphA2 has been shown to overcome acquired resistance to EGFR kinase inhibitors in lung cancer [36,37]. Although EGFR was by far the most frequently activated RTK in TET in our series, EGFR mutations are very rare [38,39,40], and the efficacy of anti-EGFR drugs appears to be very limited in non-mutated tumors [41]. Thus, studying the interaction of EGFR and Ephrin receptors in TET may merit further investigation. 

## 5. Conclusions

Using an indirect prediction model as well as direct phosphoproteomic measurements of activated RTKs, we have shown that TH and TC express multiple important sunitinib targets, thus providing a rationale for the observed (though mixed) clinical results in both entities that may help to refine further use of this drug and potentially other tyrosine kinase inhibitors. Notably, the mechanisms regulating sunitinib response in TC and TH appear different, with the striking finding that TYRO3, a potent mediator of sunitinib resistance, was overexpressed and activated in a stage-dependent manner in a high percentage of metastatic TH but not in TC. Determining TYRO3 activation in TH may thus be a novel biomarker to predict sunitinib response and may, at the same time, be a promising therapeutic target. It remains an enigmatic finding that KIT, the most constantly expressed molecule on immunohistochemistry in TC, was found to be consistently functionally inactive in 100% of the TC samples studied here and that targeting the EGFR, by far the most frequently activated RTKs in all TET, has shown minimal clinical benefit. A detailed analysis of complex RTK interactions, such as between EGFR and Ephrin receptors, may thus merit further investigation. 

## Figures and Tables

**Figure 1 cancers-14-04762-f001:**
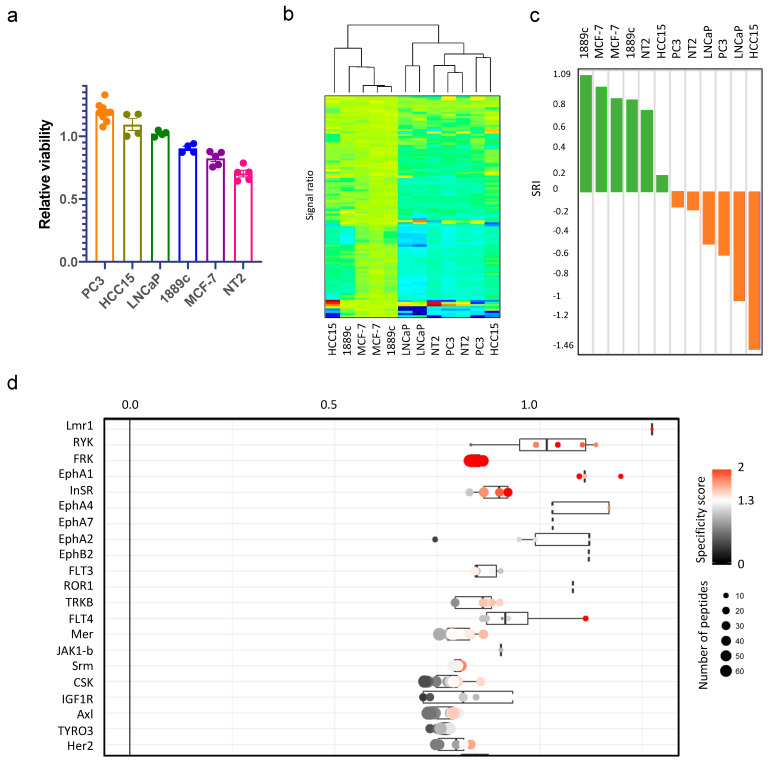
Generation of the SRI using six cell lines with different sunitinib responses. (**a**) Relative cell viability of PC3, HCC15, LNCaP, 1889c, MCF-7, and NT2 compared to untreated control after sunitinib treatment (5 µM) for 48 h. (**b**) Unsupervised hierarchical clustering of phosphorylation ratios of cell lines (sunitinib treated vs. untreated). (**c**) Waterfall plot of sunitinib-sensitive and resistant cell lines based on their SRI established by PLS-DA. Green = responding SRI, orange = resistant SRI. (**d**) Prediction of upstream tyrosine kinases responsible for sunitinib resistance. The value on the x-axis indicates the activity change of each kinase relative to untreated control. The color indicates the reliability of the prediction (specificity score). The dot size indicates the number of phosphorylated peptides on which the prediction was based.

**Figure 2 cancers-14-04762-f002:**
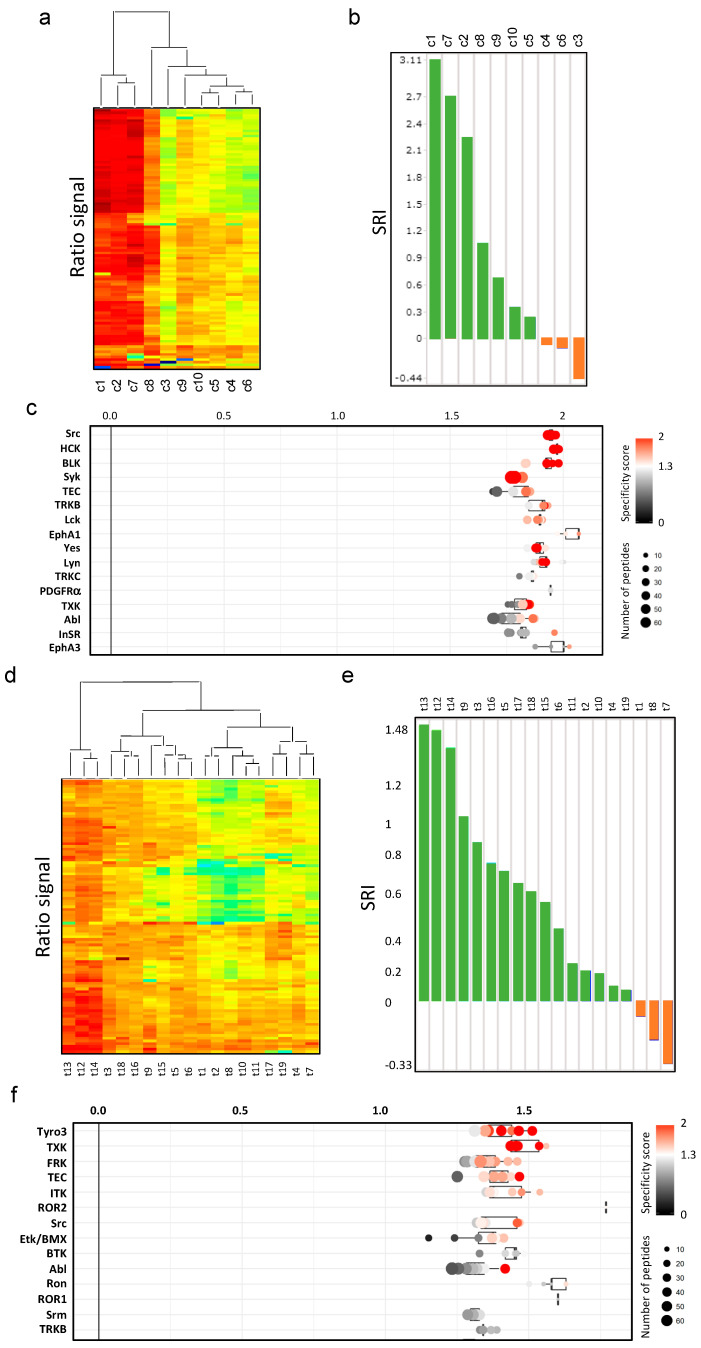
The prediction of SRI in TH and TC tissue samples and an RTK upstream prediction. (**a**) Unsupervised hierarchical clustering of sunitinib response ratios of 10 TC (c1–c10). (**b**) Predicted sunitinib response of TC based on their SRI (waterfall plot). Green = responding SRI, orange = resistant SRI. (**c**) Prediction of active upstream tyrosine kinases in TC. (**d**) Unsupervised hierarchical clustering of sunitinib response ratios of 19 TH samples (t1–t19). (**e**) Predicted SRI Waterfall plot on the 19 TH. (**f**) Prediction of active upstream tyrosine kinases in TH. The value on the x-axis indicates the activity change of each kinase relative to untreated control. The color indicates the reliability of the prediction (specificity score). The dot size indicates the number of phosphorylated peptides on which the prediction was based.

**Figure 3 cancers-14-04762-f003:**
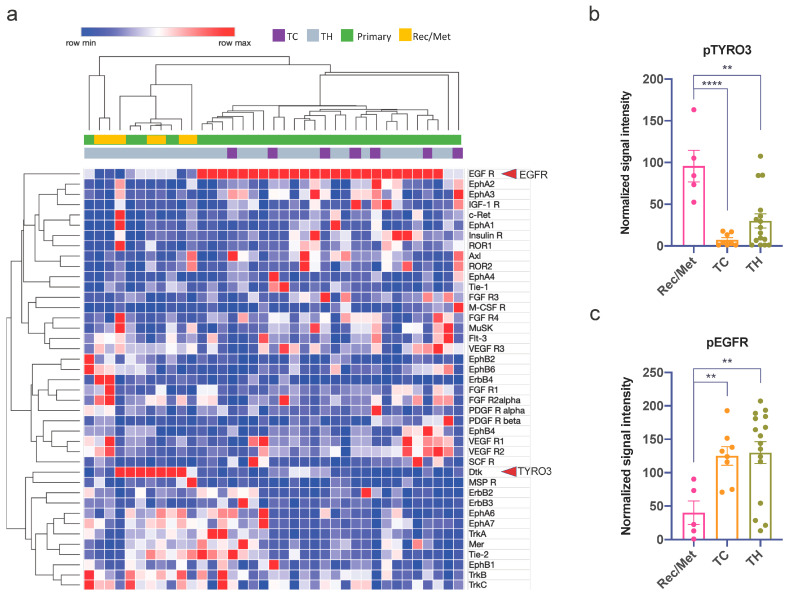
RTK activity array reveals EGFR and TYRO3/Dtk as the main targets in TET. (**a**) Heatmap of normalized RTK array dot plot signal values of 32 TH and TC shows non-overlapping signals for EGFR and TYRO3/Dtk (arrows). Metastatic TH (MET) and recurrent TH (Rec) are mainly found in the TYRO3/Dtk cluster. (**b**) Average TYRO3 signal was significantly higher in Rec/Met than in primary TH and TC. (**c**) In contrast, EGFR activation was significantly higher in primary/localized TH and TC than in Rec/Met (** *p* < 0.01, **** *p* < 0.0001).

**Figure 4 cancers-14-04762-f004:**
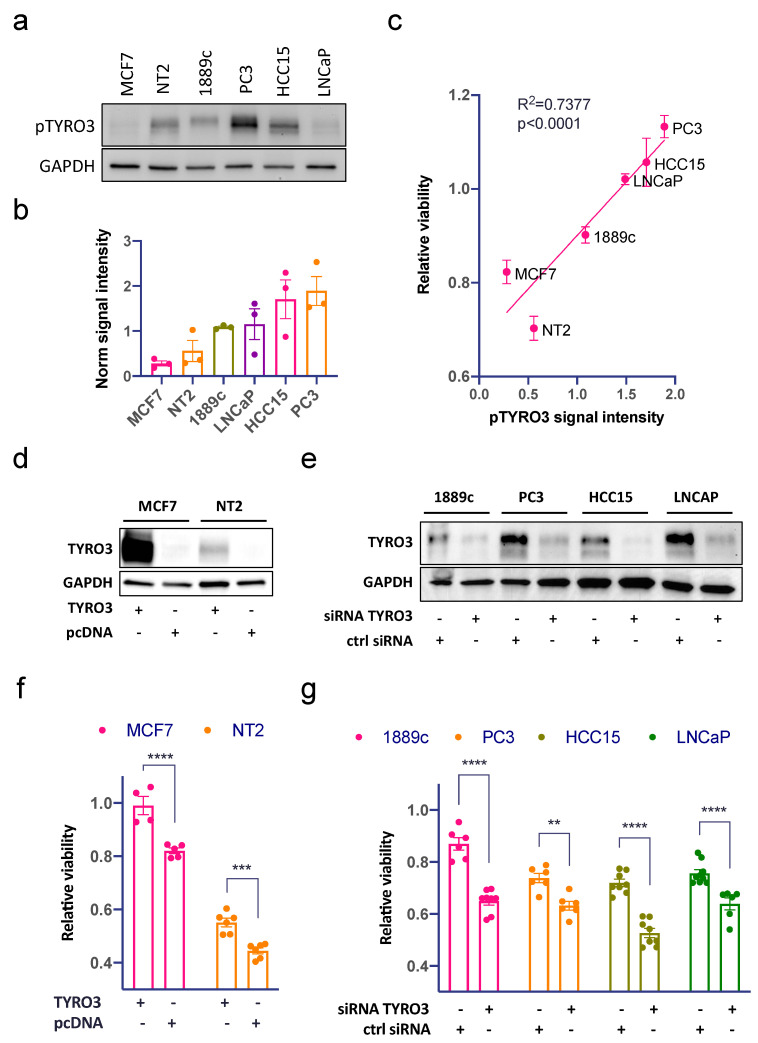
TYRO3/Dtk activity correlates with sunitinib response. (**a**) Western blot analysis of activated TYRO3/Dtk and (**b**) quantification of the TYRO3/Dtk phosphorylation signal in six cell lines (MCF-7, NT2, 1889c, PC3, HCC15, and LNCaP). (**c**) Correlation of averaged TYRO3/Dtk activity with sunitinib response in the same cell lines. (**d**) Overexpression of TYRO3/Dtk in MCF-7 and NT2 and (**f**) their relative sunitinib response compared to the native cells. (**e**) Knockdown of TYRO3/Dtk by siRNA in 1889c, PC3, HCC15, and LNCaP and (**g**) the significantly increased sunitinib sensitivity of specific siRNAs in comparison to the siRNA control (** *p* < 0.01, *** *p* < 0.001, **** *p* < 0.0001 ). The uncropped blots are shown in Appendix A.

**Table 1 cancers-14-04762-t001:** Clinicopathological parameters of TET patients.

Patients	49
female (%)	21
male (%)	51
Average age (range)	58.8 (36–84)
Thymoma	34
B1	1
B2	13
B3	20
Thymic carcinoma	15
Masaoka-Koga stage (%)	
1	0
2	23.5
3	29.4
4	47.1

## Data Availability

Normalized QC data presented in this study are available in Appendix A.

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
