# Peer review of "Phosphoproteomic Analysis Identifies TYRO3 as a Mediator of Sunitinib Resistance in Metastatic Thymomas"

_cancers, 2022, doi:10.3390/cancers14194762_

Round 1
Reviewer 1 Report
Prediction of the drug resistance will benefit to the patient's treatment. In this manuscript, the authors use the peptide microarray to measure Kinase activity profiles of sunitinib-sensitive and -resistant cell lines. They generated a sunitinib response index (SRI). 22 Protein lysates from thymomas and thymic carcinomas with spike-in of sunitinib were then classified as potential responders vs. non-responders using the same SRI classifier. Bioinformatic prediction and further experimental analysis of activated upstream kinases identified TYRO3 as a potent mediator of sunitinib resistance, specifically in metastatic thymomas. TYRO3 could serve both as a biomarker of sunitinib resistance and a potential therapeutic target that could help to tailor treatment decisions and to overcome therapy resistance in advanced thymomas and thymic carcinomas. The results are interesting and support their conclusion. Only a few issues of the figure need to be corrected.
Fig.2 a,b, d,e
Each column in the figure need to be marked.
Fig.4 s, j
The figure words are ?? but not its description.
Author Response
Fig.2 a,b, d,e
Each column in the figure needs to be marked.
We labeled all the columns in Fig. 2. We introduced for the 10 thymic carcinoma samples c1 – c10 and for the 19 thymoma samples t1 – t19). Labels were introduced in the text and the figure legend.
Fig.4 s, j
The figure words are ?? but not its description.
We did not find the mentioned Fig.4 s, j. However, in the figure legend of Fig.4 the labels for f and g were in capital letters.
Reviewer 2 Report
The paper presented for review is an interesting and unusually well prepared exploration of the relationship between sunitinib and its application to cases of thymoma and thymic cancer. The paper presents an in vitro study, carried out on a series of cell lines, which has some very interesting and valuable results.
The findings of the study are of significance, particularly in cases of thymoma, which are rather rare entities - thus any new information is of value to the community.
The paper is, as previously mentioned, of unusually high quality in terms of preparation, presentation, and use of English. The work was easy to read, follow, and understand. The extensive use of graphs, including cluster analysis and waterfall plots, demonstrated the findings in an immediately recognizable manner, allowing the reader to see the results and infer their meaning.
I have only one question to the authors regarding the experimental procedure: Transfection was by plasmid DNA or by siRNA. I believe the paper would benefit by presenting the reasoning for the use of two methods and how the methods were selected for each of the tested cell lines.
Of particular interest was the marked difference in the activation of EGFR and TYRO3/Dtk in thymoma cases, where TYRO3 was found to be significantly more phosphorylated in metastatic thymoma than in primary cases or in cases of thymic carcinoma. This finding alone is worthy of publication and, I believe, will be of great interest to readers of the journal.
Author Response
I have only one question to the authors regarding the experimental procedure: Transfection was by plasmid DNA or by siRNA. I believe the paper would benefit by presenting the reasoning for the use of two methods and how the methods were selected for each of the tested cell lines.
We thank the reviewer for his positive feedback and the helpful remarks. We changed the following section in the Material and Methods and in the Results part to clarify the selection of the cell lines and the method.
Line 174
“Cells were reverse transfected with either plasmid DNA or siRNA. Plasmid transfection was performed using the X-tremeGENE HP DNA transfection reagent (Merck, Germany) according to the manufacturer's instructions. In brief, 100 µL transfection mix containing serum-free RPMI-1640 cell culture medium, 2 µg plasmid DNA, and 2 µl transfection reagent was incubated for 15 min at room temperature and added to 4 × 105 cells in 2 ml medium directly after seeding. The TYRO3 and the control expression vectors were purchased at OriGene, USA and Thermofisher, USA (TYRO3:pCMV6-XL4, CAT#: SC108283, pcDNATM3.1(+), CAT#: V790-20). For siRNA transfection HiPerFect (Qiagen, Germany) was used according to the manufacturer's protocol. Briefly, a transfection mix of 100 µl serum-free RPMI-1640, 100 nM siRNA, and 12µl transfection reagent was incubated for 5 min and added to 4 × 105 cells in 2.3 ml medium and incubated for 24 hours before treatment. siRNAs were purchased from Qiagen, Germany (SI00288344, SI00288351).“
Line 325
“Testing pTYRO3/Dtk activity in the six cell lines, we detected variable signals of activated TYRO3 (Fig. 4a and b). The pTYRO3/Dtk signal intensity was statistically significantly correlated with sunitinib response (Fig. 4c). To further validate the func-tional importance of TYRO3/Dtk, we used an expression vector (pCMV6-XL4, Origene, CAT#: SC108283) to transiently overexpress TYRO3/Dtk in the two cell lines with the lowest constitutive TYRO3 activation and the best response to sunitinib (MCF-7, and NT2). This significantly increased the resistance against sunitinib in both cell lines (Fig. 4d and f). Vice versa, in the four cell lines with strong constitutive activation and a lesser response to sunitinib (1889c, PC3, HCC15, and LNCaP), gene silencing of TY-RO3/Dtk with two individual siRNAs (SI00288344, SI00288351, Qiagen) resulted in sig-nificantly more tumor cell killing upon exposure to sunitinib (Fig. 4e and g).”